# Synthesis of Propylene-*co*-Styrenic Monomer Copolymers via Arylation of Chlorinated PP and Their Compatibilization for PP/PS Blend

**DOI:** 10.3390/polym11010157

**Published:** 2019-01-17

**Authors:** Xiangming Fu, Xijun Liu, Chunyu Zhang, Heng Liu, Yanming Hu, Xuequan Zhang

**Affiliations:** 1College of Materials Science and Engineering, Qiqihar University, Qiqihar 161006, China; xmfu@ciac.ac.cn; 2CAS Key Laboratory of High-Performance Synthetic Rubber and its Composite Materials, Institute of Applied Chemistry, Chinese Academy of Sciences, Changchun 130022, China; hengliu@ciac.ac.cn (H.L.); ymhu@ciac.ac.cn (Y.H.); xqzhang@ciac.ac.cn (X.Z.)

**Keywords:** polypropylene, chlorinated polypropylene, polystyrene, polymer blend, compatibilization

## Abstract

A series of propylene-*co*-styrenic monomer copolymers were synthesized using the Friedel–Crafts alkylation reaction between chlorinated PP and substituted benzene, and the effects of these copolymers on a PP/PS (80/20) blend were investigated by using the impact test, morphology observation, thermo- and dynamic mechanical analysis, and rheology measurements. The results showed that the compatibilization efficiency varied as the variation of the substitute on the benzene ring of the styrenic monomer unit was incorporated in the PP chain in an order of methyl > ethyl > methoxyl. The copolymers bearing a crystalline isotactic polypropylene chain sequence and rubbery propylene-*co*-styrene-like unit chain segments may prepossess imaginable applications, giving an example for the synthesis and applications of PP-based copolymers, initiating a new way to broaden the polyolefin-based material family.

## 1. Introduction

As is well known, the importance of polypropylene (PP) cannot be over complimented in the modern world [1,2]. With great effort over decades, substantial progress has been made to meliorate the weaknesses of this material, but these have still been far from satisfactory until now. Modification in the polymerization process has proven to be successful in the preparation of PP in reactor alloys [3]; whereas, due to the limitations of the intrinsic nature of the Ziegler–Natta catalyst, the preparation of PP with a branched structure, and that with non-α-olefinic monomers such as styrenic and polar monomers incorporated in the main chain, tends to be extremely difficult by directly using the polymerization process. Therefore, post-polymerization reactions are considered to be more practical and have been extensively studied [4].

Chlorination of polyolefin by using the reaction of polyolefin nascent particles with chlorine gas in water media is an easy and economical way to obtain functionalized polyolefin materials due to the inexpensive and excess chlorine gas resource resulting from the chlor–alkali industry. Chlorinated polyethylene (CPE) and chlorinated polyethylene rubber (CM) have been commercialized in large capacity since the materials have been endowed with excellent weather fastness, flame retardancy, and elastomeric properties by chlorination [5]. However, chlorinated PP (CPP) still has problems to be a useful material since the thermo-instable C–Cl bond cannot be sustained at the processing temperature of crystalline PP chain segments (normally, over 170 °C), giving rise to the release of unpleasant corrosive HCl gas accompanied by severe molecular weight decline [6]. From our viewpoint, the C–Cl bond can be employed as a reaction site for organic substitution reaction to convert the chlorine atom to a thermally-stable organic group and to open up a way to enrich the polyolefin material family and expand the applications as an expectation.

Herein, we attempted to synthesize polypropylene with styrenic monomer units incorporated in the main chain by using the Friedel–Crafts alkylation reaction between CPP and substituted benzene as raw materials together with the Lewis acid as the catalyst. Although the Friedel–Crafts alkylation reaction using CPP as a component has been reported for the preparation of in situ compatibilized PP-based polymer blends [7], it has not been reported yet as a pathway for the synthesis of polypropylene with styrenic monomers incorporated in the backbone and utilization as a compatibilizer for PP/PS blends. As a combination of two major polymeric materials, PP/PS blends are undoubtedly of great scientific and commercial interest. Although various commercial compatibilizers such as PP graft maleic anhydride (PP-*g*-MAH), Polystyrene-*b*-poly(ethylene-*r*-butylene)-*b*-polystyrene (SEBS), Surlyn, etc. have been reported for compatibilization of PP/PS blends [8,9], the compatibilization efficiencies of those compatibilizers cannot be well complimented due to insufficient structural similarity. It is interesting from a scientific aspect that a well-defined polypropylene-*graft*-polystyrene synthesized by a two-step polymerization process and used as compatibilizer for PP/PS blend have been reported [10].

In this work, a series of propylene-*co*-styrenic monomer copolymers were synthesized by using the Friedel–Crafts alkylation reaction, and those copolymers as a compatibilizer for the PP/PS blend were investigated by using the impact test, morphology observation, dynamic mechanical analysis, thermo- and rheological behavior measurements.

## 2. Experimental Section

### 2.1. Materials

Chlorinated polypropylene (CPP) with chlorine content of 10.1 wt.% and weight average molecular weight (*M*_w_) of 97,500 g/mol was supplied by ZHONGXU Polyolefin Co., Ltd. (Suqian, China). PP, Grade 5D46, melt flow index, 3.4 g/10 min (230 °C/2.16 kg), was supplied by CNPC Fushun Petrochemical Company (Fushun, China). PS, Grade GPPS123P, melt flow index, 8 g/10 min (200 °C/5.00 kg), was supplied by SECCO Petrochemical Co., Ltd. (Shanghai, China). Polystyrene-*b*-poly(ethylene-*r*-butylene)-*b*-polystyrene (SEBS), weight average molecular weight (*M*_w_), 118,000 g/mol, was purchased from Sigma Aldrich. Antioxidant 1010 was supplied by JIYI Chemical Co., Ltd. (Beijing, China). Benzene, toluene, ethyl benzene, methoxyl benzene, decalin, aluminum chloride, etc., were of analytical grade and treated according to standard procedures before use.

### 2.2. Synthesis of Propylene-co-Styrenic Monomer Copolymers

The molar ratio of CPP (in Cl), aluminum chloride (in Al), and substituted benzene (benzene, toluene, ethylbenzene, methoxyl benzene) was set as 1:1:1. Aluminum chloride was added to a 70 °C CPP decalin homogeneous solution, followed by dropwise-injected benzene, toluene, ethylbenzene, and methoxyl-benzene, respectively, and stirred for 8 h. The copolymers were individually collected by precipitation in excess ethanol and washed with ethanol several times, then dried in vacuum at 50 °C to constant weight. For convenience, the copolymers obtained were termed as P0, P1, P2, and P3 in the order of benzene, toluene, ethylbenzene, and methoxyl benzene as the alkylation reagent.

### 2.3. Preparation of Blends

PP/PS (80/20) blends were prepared first by premixing using a high-speed mixer, then by melt-mixing using a WLG10G double-screw mixing system (XINSHUO precision machinery Co., Ltd., Shanghai, China) at 210 °C with a rotor speed of 60 rpm for 5 min, and finally, injection molding at 190 °C using a WZS10D injection molding machine (XINSHUO precision machinery Co., Ltd., Shanghai, China). The binary blend and that compatibilized with varied amounts of propylene-*co*-styrenic monomer copolymers and SEBS as references were individually obtained as testing specimens.

### 2.4. Instrumentation

#### 2.4.1. Nuclear Magnetic Resonance

^1^H-NMR measurements were carried out on a Varian Unity 400 spectrometer (Varian, Palo Alto, CA, USA) at room temperature with CDCl_3_ as the solvent.

#### 2.4.2. Impact Test

The notched impact strengths of all samples were tested at room temperature by using a JJ-20 memory impact testing machine (Changchun Intelligent, Changchun, China). The sample dimensions were 78 × 9 × 4 mm^3^.

#### 2.4.3. Morphology Observation

Morphologies of the blends were observed by using an S-4300 scanning electron microscopy (SEM) (Meiji, Japan). The experiments were done at an activation voltage of 15 kV under high vacuum. The impact fractured surfaces were coated with thin layers of gold before observation. Some specimens were etched with butanone in case of need before gold sputtering.

#### 2.4.4. Dynamic Mechanical Analysis

Dynamic mechanical analysis (DMA) was examined using a DMA+450 dynamic mechanical analyzer (01DB-METRAVIB, Netherlands, French). Rectangular specimens with dimensions of 60 × 10 × 4 mm^3^ were tested by using a dual cantilever clamp at a dynamic frequency of 1 Hz in temperatures ranging from −50–140 °C and a heating rate of 3 °C/min.

#### 2.4.5. Thermoanalysis

Melting and crystallization properties were examined by using a Q20 (TA, Newcastle, DE, USA) differential scanning calorimeter (DSC) in a dry nitrogen atmosphere (the outlet pressure was about 0.1 MPa). Approximately 10 mg of samples were heated to 200 °C and kept for 5 min to eliminate heat history, and then data were recorded while samples were cooled down to 50 °C and heated again to 200 °C at a scanning rate of 10 °C/min.

Crystallinity of PP in the blends was calculated according to the normalized fusion enthalpy of the samples, weight percent of PP in the blend, and fusion enthalpy of PP in 100% crystallinity, i.e., 207.1 J/g [11].

#### 2.4.6. Rheology Measurement

Rheology measurements were conducted on an ARES rheometer (TA, Newcastle, DE, USA) with a parallel plate geometry of 25 mm in diameter under nitrogen. The tests were performed at an isothermal frequency ranging from 100–0.01 rad/s for all samples at 190 °C at a strain of 5%.

## 3. Results

### 3.1. Nuclear Magnetic Resonance

Propylene-*co*-styrenic monomer copolymers were synthesized via the Friedel–Crafts alkylation reaction between chlorinated polypropylene and substituted benzene using Lewis acid as the catalyst. As shown in Figure 1, the chain structure of chlorinated polypropylene is well characterized [12], but fairly complicated, consisting of chain units with isolated chlorine atoms (83.4%) including those at the tertiary (65.9%) and secondary (17.5%) carbon, neighboring chlorine atoms (6.4%), and small amounts of chlorine methane (10.1%). During the Friedel–Crafts alkylation reaction, the chlorine at different carbons causes the formation of different carbocation and gives rise to different chain units incorporated in the PP chain of the resulting copolymer when further reacted with substituted benzene [13,14]. Since the Friedel–Crafts alkylation reaction undergoes a carbocation mechanism, the carbocation intermediate also causes alternative C–C bond cleavage besides reacting with the substituted benzene, resulting in a decrease in the molecular weight of the copolymer. As the first priority given its structural similarity to the styrene unit in PS, benzene used as an alkylation reagent in the Friedel–Crafts reaction in our first trial proved unsuccessful due to its too low molecular weight, which was less than 10,000 g/mol of the resulting copolymer (P0), caused by severe cationic chain scission. Henceforth, substituted benzene with an electron donating group, i.e., methyl, ethyl, and methoxyl groups, was used instead of benzene in the Friedel–Crafts alkylation reaction, and the results showed that the electron-donating substitute could effectively suppress carbocation chain scission as the molecular weight of the copolymer increased with increasing electron-donating potentiality in the order of methoxyl > methyl > ethyl (Table 1).

By using the Friedel–Crafts alkylation reaction, poly(propylene-*co*-methyl-styrene), poly(propylene-*co*-ethyl styrene), and poly(propylene-*co*-methoxyl styrene) were synthesized accompanied by molecular weight decline due to the carbocation chain scission side reaction. As shown in Figure 2, the reaction occurred predominately on the para-position of substituted benzene, most likely due to more steric hindrance on that of the ortho-position, and this was the reason that there was no multi-alkylation on the same benzene ring observed in Figure 2, which could cause cross-linking of the polymer chain. The content of the remnant chlorine atom on the polymer chain is too low to be visible in the NMR spectra, and the trace could only be detected by ICP analysis (Table 1), suggesting that the chlorine atoms were almost consumed by the Friedel–Crafts alkylation and chain scission reactions (as shown in Scheme 1) [15].

Due to the fair complexity of the chain structure of CPP, carbocation rearrangement, and chain scission, the precise chain structure of the resulting copolymers cannot be fully defined yet. Nonetheless, the alkylation of substituted benzene on the PP chain turning into styrenic monomer units was clearly proven, as shown in Figure 2, and the contents of the styrenic units in the aromatic ring/100 C in corresponding copolymers were calculated according to the peak area at 2.42 ppm attributed to CH_3_ in methyl-styrene; 2.62 ppm was assigned to CH_2_ in the ethyl group of ethyl styrene; a 3.79 ppm peak area at 2.42 ppm was attributed to CH_3_ in methyl-styrene; 2.62 ppm was assigned to CH_2_ in the ethyl group of ethyl styrene; and 3.79 ppm was assigned to CH_3_ in the methoxyl styrene.

### 3.2. Mechanical Properties

The binary blend of PP/PS (80/20) and that compatibilized by using propylene-*co*-styrenic monomer copolymers synthesized above, and together by using SEBS as a reference were prepared by melt mixing. Figure 3 shows the impact strength of the blend and that compatibilized by using copolymers with different substitutes on the benzene ring and SEBS. The results showed that all the compatibilized blends had a better impact strength than that of the binary PP/PS blend (4.1 kJ/m^2^), implying enhanced interfacial adhesion between the dispersed PS domain and PP matrix to the extent depending on the compatibilizer used [16]. In comparison, the blend compatibilized with the copolymer with methylstyrene units (P1) possessed a better impact property than with those with ethylstyrene and methoxylstyrene units (P2, P3) and was even better than those of SEBS, which is commonly used in polyolefin-based blending materials. The difference in compatibilization effectiveness can probably be attributed to the intrinsic chain structure of the individual compatibilizer, since the more flexible ethyl and methoxyl groups in P2 and P3 than that of the methyl group in P1 may weaken the π–π interaction [17] between the benzene rings of PS and that of the compatibilizer, so lowering the compatibilization efficiency, giving rise to the worse impact property of the blends. In the case of SEBS, though it contained the PS block, the poor compatibilization of its EB block with PP [18] limited its compatibilization effectiveness in the PP/PS blend system. From this point of view, the better impact strength than that of the binary PP/PS blend can mostly be attributed to the toughening effects of its intrinsic elastomeric nature [19]. As commonly seen, the impact strength of the blend with each compatibilizer fluctuated as the variation of the compatibilizer amount and the maximum impact strength of the blends varied, as well with the compatibilizer used [20,21].

### 3.3. Morphology Observation

SEM micrographs of the binary blend of PP/PS (80/20) and ternary blends with different compatibilizers are shown in Figure 4. For clarity in the image, PS dispersed domains on the fractured surface were etched by using butanone. The statistical phase diameter of the blend system from the equal area circle diameter measurement is tabulated in Table 2. The binary blend of PP/PS (80/20) showed the typical morphology of an immiscible blend system (Figure 4a) [22,23,24] with PS domains up to 5 µm in diameter dispersed in the PP matrix. The ternary blends showed a much smaller dispersed domain size, proving that compatibilization effectiveness more or less depended on the compatibilizer used. As revealed by the statistical phase diameter data in Table 2, P1 with methylstyrene units incorporated showed better compatibilization efficiency in the blend system than that with ethylstyrene units (P2) and was much better than that with methoxylstyrene units (P3) and SEBS, which was consistent with the impact testing results. Relatively poor compatibilization efficiency of P3 resulted from the weak compatibility of its methoxylstyrene units with PS due to frustrated π–π interaction between the intermolecular benzene rings caused by its flexible methoxyl group. In contrast, the poor compatibility of the EB block of SEBS with PP was responsible for its poor behavior in the PP/PS blend system [25,26,27].

Figure 5 shows the SEM images of the PP/PS blends with varied amounts of P1 as the compatibilizer. With the increment of P1 content, the dispersed PS domain size decreased to the minimum at 2 wt.% and then increased again. As normally seen for most polymer blends, there exists the optimal compatibilizer content, agglomerating itself to new domains instead of diffusion at the interfacial area, which is probably the reason for what occurs at higher compatibilizer content [28].

### 3.4. Dynamic Mechanical Analysis

The *T*_g_ (glass transition temperature) of polymer blends may provide information referring to the compatibility between the components, which shifts either inward or outward in most cases. As shown in Figure 6, the *T*_g_ of the PS blend shifted inward to 108.2 °C from 110.5 °C in the binary PP/PS blend when P1 was used as the compatibilizer, but shifted outward when P2 and P3 were employed. The more flexible non-crystalline PP chain segments in the vicinity of methylstyrene units of P1 enhanced the mobility of the PS chains [29], so lowering the *T*_g_ of the PS dispersed phase, which is strong evidence of the better compatibilization efficiency of P1 than those of P2 and P3 [30,31].

### 3.5. Thermoanalysis

In Figure 7, the crystallization temperature (*T*_c_) and crystallinity of PP in ternary PP/PS blends increased obviously in comparison with that of the binary blend when P1 and P2 were used as compatibilizers, but did not change much when P3 and SEBS were used. In contrast to the use of P3 and SEBS as the compatibilizer, the better compatibilization of P1 may lead to greater intermolecular diffusion of the components [32,33,34], enabling the PP chains to crystallize more easily by heterogeneous nucleation, and consequently, fast nucleating and more nuclei caused higher *T*_c_ (Figure 7a) and higher crystallinity (Figure 7b) [35,36,37,38].

### 3.6. Rheology Measurement

Figure 8 shows the rheological behaviors of the binary PP/PS (80/20) blend and those compatibilized using 2 wt.% of compatibilizers, together with virgin PP and PS as references. The low viscosity of PP in the full shear rate range resulted from its linear chain structure and low intermolecular interaction in the melt, which is responsible for the poor melt strength. In contrast, PS showed a much high melt viscosity as the much stronger inter- and intra-molecular interaction resulting from π–π stacking between the benzene rings pendant from the main chain endows PS with a superior melt strength. The binary PP/PS blend showed some higher melt viscosity than that of PP resulting from 20 wt.% PS being blended. Surprisingly, the PP/PS (80/20) blend with 2 wt.% of P1 showed a melt viscosity much closer to that of PS, even though PP was the dominating component and matrix of the blend. The much-improved intermolecular interaction of PP chains and between PP and PS in the blend can reasonably be attributed to the compatibilization of P1 with methylstyrene units incorporated randomly along the PP chain in the way of π–π stacking between the benzene rings of PS and P1 [39]. It is worth emphasizing from the discussion above that the melt strength of PP could be significantly improved by blending polymeric materials of high melt strength together with proper compatibilizers, thus opening up a new approach for the preparation of high melt strength PP in an expedient way.

## 4. Conclusions

A series of propylene-*co*-styrenic monomer copolymers was synthesized using the Friedel–Crafts alkylation reaction of chlorinated PP and substituted benzene and employed as compatibilizers to the PP/PS (80/20) blend. The results showed that the compatibilization efficiency varied with the variation of the substitute on the benzene ring of styrenic monomer unit incorporated in copolymers in an order of methyl > ethyl > methoxyl, most likely resulting from their differentiated effects on π–π stacking between the benzene rings of PS and the compatibilizer. The good compatibilization of the propylene-*co*-methyl-styrene copolymer was revealed by the improved PS domain dispersion, inward shift of *T*_g_ of PS in the blend, and further evidenced by the ameliorated impact strength, demonstrating the significance of structural similarity between the compatibilizer and the blended components. Additionally, the melt strength of PP could be much improved, nearing that of virgin PS by blending only 20% PS together with a 2 wt.% copolymer, guiding a new way to produce high melt strength PP. In the end, the current work shows an example of the synthesis of PP-based copolymers using an organic substitution reaction with chlorinated PP as the starting material, and it is expected that this can provide a starting point for novel approaches for synthesizing polyolefin-based copolymers and, consequently, the enlarging polyolefin material family.

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
