# Peer review of "Synthesis of Propylene-co-Styrenic Monomer Copolymers via Arylation of Chlorinated PP and Their Compatibilization for PP/PS Blend"

_polymers, 2019, doi:10.3390/polym11010157_

Round 1

Reviewer 1 Report

This paper describes arylation of chlorinated polypropylene with Friedel-Crafts type reaction and application of the resulting polymer to the compatibilizer of PP/PS composit. The prepared arylated polymer lendered higher impact strength on the PP/PS blend than block copolymer (SEBS) does.
The reviewer feels there are some concerning about the characterization of the polymer, which makes difficult to further discuss about the origin of sucessful compatibilization:

1. In P1 to P3 the conversion of chloride to aromatic ring seems to be almost quantitative, which means that approximately 10 mol% of styrene moiety was randomly incorporated into PP sequence. Usually such a comonomer incorporation would extinguish the crystallinity of PP, but the results were saying that melting temperature was going up. Pleases discuss it.
2. Considering the mechanism of Fridel-Crafts reaction, stereoregularity on some tertially carbons would be changed. In addition, only a part of aromatic signals were assigned with 1H NMR although the authors denied the multialkylation on substituted benzene. To make them clear, 13C NMR assignment should be shown at least.
3. The starting chlorinated PP contains significant amount of C=C double bonds but the resulting polymer does not. This is not only because of less frequent chain scission but some functionalization which C=C bond participating in. Please raise the possible reaction pathways.
4. Related to the comment 3, aren't there any possibility that the main chain of the resulting polymer contains cyclic part?

Author Response

Reply to the comments by Reviewer 1

Thank you very much for your comments. We carefully read your comments, and our revisions and replies are as follows:

1. In P1 to P3 the conversion of chloride to aromatic ring seems to be almost quantitative, which means that approximately 10 mol% of styrene moiety was randomly incorporated into PP sequence. Usually such a comonomer incorporation would extinguish the crystallinity of PP, but the results were saying that melting temperature was going up. Pleases discuss it.

Yes, as the reviewer mentioned, the crystallization of PP was already substantially affected by chlorination, the conversion of chloride to aromatic ring did not alter the sequence length of crystalline PP chain and crystallinity as results. The slight increase of Tm from 140.5℃ of CPP to 144.4, 144.7 and 151.7℃ of P1, P2 and P3, respectively, could be caused by stronger nucleation of aromatic unit than that of chloride substituted unit. The aromatic ring containing nucleating reagent are usually efficient for polyolefin (e.g., Polymer, 1970, 11, 309-332 and Journal of Applied Polymer Science, 2006, 100, 4868–4874).

2. Considering the mechanism of Fridel-Crafts reaction, stereoregularity on some tertially carbons would be changed. In addition, only a part of aromatic signals were assigned with 1H NMR although the authors denied the multialkylation on substituted benzene. To make them clear, 13C NMR assignment should be shown at least.

Yes, as the reviewer’s comment, the stereoregularity on some tertiary carbons should be changed when the orbital of carbon atoms changes from sp3 hybridization to planar carbocation sp2 hybridization.

In addition, the 13C-NMR spectra were supplemented. Although the signals assigned to benzene ring is despondently not recognizable, we still consider that multi alkylation on substituted benzene occurs in low possibility in the presence of excess substituted benzene.

13C-NMR spectrum of P1

13C-NMR spectrum of P2

13C-NMR spectrum of P3

3. The starting chlorinated PP contains significant amount of C=C double bonds but the resulting polymer does not. This is not only because of less frequent chain scission but some functionalization which C=C bond participating in. Please raise the possible reaction pathways. 

C=C double bonds in chlorinated PP formed in the chlorination process, and disappeared in Fridel-Crafts reaction process. Possibly, reaction with carbocation forming cyclic units and participation in Fridel-Crafts reaction as alkylation reagent are main reasons for disappearance of C=C double bonds. For more clarity, we carried out a contrast experiment, i.e., at the same reaction conditions as performed in Fridel-Crafts reaction as mentioned in the manuscript besides no alkylation reagent was used, the C=C double bonds in CPP disappeared as well after reaction as shown in H-NMR below, indicating cyclization reactions are responsible for the disappearance of C=C double bonds.

4. Related to the comment 3, aren't there any possibility that the main chain of the resulting polymer contains cyclic part?

Sure, the resulting polymer should contain cyclic part, as discussed in comment 3, the cyclization reactions were responsible for the disappearance of C=C double bonds in resulting polymer.   

Reviewer 2 Report

The title is misleading in unacceptable measure, as a matter of fact the products of the described synthesis are chlorinated polypropylenes with occasional aromatic ring added to the backbone (no styrene at all). Some copolymers of i-PP with polystyrene grafts were reported in the literature and investigated as compatibilizers [see e.g. L. Caporaso, N. Iudici, L. Oliva “Synthesis of well-defined polypropylene-graft-polystyrene and relationship between structure and the ability to compatibilize the polymeric blends” Macromolecules38, 4894-4900 (2005)]. A comparison or at least a citation should be necessary.

In Table 1 one can observe the relevant increase of the melting temperature of P3 with respect the starting material CPP. It seems surprising such an experimental evidence (10 degrees of difference) and should be discussed.

In Figure 1 is reported the H NMR spectrum of the chlorinated polypropylene with careful assignment of the peaks, but how such an assignment was performed? It is original or was previously reported? The indication of reference or the discussion, as is the case, must be reported.

The discussion around the thermal behavior is, in my opinion, feeble. The curves of figure 7 are arbitrary and actually the experimental points representative of the melting temperatures are constant with increasing the compatibilizer %, oscillating in the range of the uncertainty of the measurement. Similarly one could state for the crystallization degree. Finally the SEM micrographs are scarcely convincing of the compatibilization effect.

Finally as further serious flaw the work is lacking of experiments using the chlorinated polypropylene as compatibilizer, the blank tests necessary to support the usefulness of the described reactions of addition of aromatic rings.

Author Response

Reply to the comments by Reviewer 2

Thank you very much for your comments. We carefully read your comments, and our revisions and replies are as follows:

1. The title is misleading in unacceptable measure, as a matter of fact the products of the described synthesis are chlorinated polypropylenes with occasional aromatic ring added to the backbone (no styrene at all). Some copolymers of i-PP with polystyrene grafts were reported in the literature and investigated as compatibilizers [see e.g. L. Caporaso, N. Iudici, L. Oliva “Synthesis of well-defined polypropylene-graft-polystyrene and relationship between structure and the ability to compatibilize the polymeric blends” Macromolecules38, 4894-4900 (2005)]. A comparison or at least a citation should be necessary.

Yes, we have modified the title “Synthesis of polypropylene-co-styrenic monomer copolymers and their compatibilization for PP/PS blend” to “Synthesis of polypropylene-co-styrenic monomer copolymers via arylation of chlorinated PP and their compatibilization for PP/PS blend”.

Thank you for your kind comment, the paper, “Macromolecules, 2005, 38, 4894-4900.”, have been cited in Reference part. In addition, a sentence was added in page 2, line 51-53, “It is interesting from scientific aspect, a well-defined polypropylene-graft-polystyrene synthesized by a two-step polymerization process and used as compatibilizer for PP/PS blend have been reported [10]”.

2. In Table 1 one can observe the relevant increase of the melting temperature of P3 with respect the starting material CPP. It seems surprising such an experimental evidence (10 degrees of difference) and should be discussed.

The increase of Tm from 140.5℃ of CPP to 151.7℃ of P3 could be caused by stronger nucleation of aromatic unit than that of chloride substituted unit. The aromatic ring containing nucleating reagent are usually efficient for polyolefin [e.g., Polymer, 1970, 11, 309-332 and Journal of Applied Polymer Science, 2006, 100, 4868–4874]. In addition, Higher Tm of P3 than that of P1 and P2 is attributed to higher molecular weight of P3 than that of P1 and P2. Polarized microscopy photographs of CPP (a), P1 (b) and P3 (c) are given below.

3. In Figure 1 is reported the H NMR spectrum of the chlorinated polypropylene with careful assignment of the peaks, but how such an assignment was performed? It is original or was previously reported? The indication of reference or the discussion, as is the case, must be reported.

At first, we sincerely apologize to reviewer about the careless on assignment of H-NMR peaks. We corrected the assignment of peaks in Figure 1 and calculation of the percentage of chlorinated polypropylene unit was revised in page 3, line 118-120.

The Table below is the assignments of peaks in CPP as reference. The reference paper “J. Polym. Sci.: Polym. Chem. Ed. 1974, 12, 1653-1669.” was cited in manuscript (page 3, line 118) and added in Reference part.

Figure 1. 1H-NMR spectrum of chlorinated polypropylene

δ, ppm

Structure assignment *

Position of carbons

measured

calculated

a

1.03

1.03

γ

b

1.13

1.12

β

c

3.42

3.42

α

d

3.55

3.54

αγ

e

3.67

3.74

α

f

3.88

3.94

α

g

4.17

4.12

α

h

1.96

1.92

γ

i

2.09

2.10

γγ

j

2.26

2.24

β

k

2.42

2.33

βγ

l

2.58

2.54

ββ

4. The discussion around the thermal behavior is, in my opinion, feeble. The curves of figure 7 are arbitrary and actually the experimental points representative of the melting temperatures are constant with increasing the compatibilizer %, oscillating in the range of the uncertainty of the measurement. Similarly one could state for the crystallization degree.

Yes, we accepted, the Tm and Xc of blends are almost constant with increasing the compatibilizer. We have made the corresponding modification in Figure 7.

5. Finally the SEM micrographs are scarcely convincing of the compatibilization effect.

Yes, we agree the SEM images are not so remarkable, but still make sense at low dosage of PS (20wt%) in PP/PS blend.

6. Finally as further serious flaw the work is lacking of experiments using the chlorinated polypropylene as compatibilizer, the blank tests necessary to support the usefulness of the described reactions of addition of aromatic rings.

Yes, we agree, but the experiment cannot be carried out since the thermo-instable C–Cl bond cannot be sustained at the processing temperature of crystalline PP chain segments, giving rise to the release of unpleasant corrosive HCl gas accompanied by severe molecular weight decline.

Round 2

Reviewer 1 Report

The authors replied all the concernings the reviewers made in the previous version. Now the reviewer thinks manuscript can be published as is submitted.

Reviewer 2 Report

The manuscript was emended of some of its major weakness points and now can be published, in my opinion